# Evaluation of the Effect of mRNA and Inactivated SARS-CoV-2 Vaccines on the Levels of Cytokines IL-2, IFN-γ, and Anti-RBD Spike SARS-CoV-2 Antibodies in People Living with HIV (PLHIV)

**DOI:** 10.3390/biomedicines12092115

**Published:** 2024-09-18

**Authors:** Amanah Amanah, Ibnu Agus Ariyanto, Budiman Bela, Risnandya Primanagara, Pratiwi Sudarmono

**Affiliations:** 1Doctoral Program in Biomedical Sciences, Faculty of Medicine, Universitas Indonesia, Jakarta 10430, Indonesia; 2Department of Immunology, Faculty of Medicine, Swadaya Gunung Jati University, Cirebon 45132, Indonesia; 3Department of Microbiology, Faculty of Medicine, Universitas Indonesia, Jakarta 10430, Indonesia; ibnu.agus71@ui.ac.id (I.A.A.); budiman.bela@ui.ac.id (B.B.); 4Virology and Cancer Pathobiology Research Center, Universitas Indonesia, Jakarta 10430, Indonesia; 5Department of Bioinformatics, Faculty of Medicine, Swadaya Gunung Jati University, Cirebon 45132, Indonesia; primanagara@gmail.com

**Keywords:** HIV, SARS-CoV-2, mRNA vaccine, inactivated vaccine, anti-SARS-CoV-2 RBD IgG

## Abstract

The safety of the mRNA and inactivated SARS-CoV-2 vaccine has been demonstrated for people living with HIV (PLHIV). However, vaccine studies in PLHIV are limited, and there is a gap in which vaccine type provides the best response in PLHIV. Thus, PLHIV may benefit from mRNA vaccine types compared to inactivated vaccines. This study aims to assess the immune responses to vaccination by measuring specific antibodies (IgG) targeting the receptor binding sites (RBDs) of the SARS-CoV-2 virus and the levels of IL-2 and IFN-γ in plasma. A total of 41 PLHIV who regularly take antiretroviral therapy (ART) over a period of six months, along with 31 individuals in a healthy control group (HC), were administered either two mRNA or inactivated vaccines. Data regarding demographics and clinical information were gathered from the medical records. An analysis was conducted on the neutralisation antibody IgG specific to RBD using the chemiluminescence microparticle assay (CMIA). The levels of IL-2 and IFN-γ were quantified using the Luminex assay method from plasma samples. Data were collected in the laboratory 28 days after each vaccination. After the first vaccination, the level of anti-SARS-CoV-2 RBD IgG was higher in PLHIV who received the mRNA vaccines than those who received inactivated vaccines (*p* = 0.006). The levels of mRNA in the PLHIV group showed a significant correlation with IL-2 and IFN-γ after the second vaccination (*r* = 0.51, *p* = 0.0035; *r* = 0.68, *p* = 0.002). The group of PLHIV who received the inactivated vaccine showed increased IL-2 and IFN-γ after the initial vaccination, compared to PLHIV who received the mRNA vaccine (*p* = 0.04; *p* = 0.08). Administering a two-dose vaccination is essential to increase the levels of neutralising antibodies significantly (*p* = 0.013) in PLHIV who have received inactivated vaccines; further study is needed to make this a recommendation. The responses observed after vaccination in PLHIV were not affected by their CD4 cell counts. PLHIV showed higher levels of SARS-CoV-2 IgG and increased IL-2 and IFN-γ levels. Our study encourages SARS-CoV-2 vaccination in PLHIV regardless of its CD4 cell counts. Furthermore, the mRNA vaccine may give robust high antibody responses in PLHIV.

## 1. Introduction

HIV belongs to the genus Lentivirus and is a member of the Retroviridae family known for its extended incubation and illness duration [1]. Viral infection primarily targets CD4^+^ T-cells and macrophages when the number of CD4 T-cells drops by 200 cells/µL; people living with HIV (PLHIV) then become susceptible to opportunistic infections, including SARS-CoV-2 [2]. Individuals with a compromised immune system often experience extended periods of SARS-CoV-2 infection and exhibit genetic alterations in the spike protein of the virus [2]. These changes are associated with the virus’s ability to evade neutralising antibodies (NAbs) [2]. Immunocompromised patients have a greater risk of becoming infected and are vulnerable to severe clinical outcomes [3,4]. Therefore, SARS-CoV-2 vaccinations are prioritised for these groups to prevent adverse conditions [5,6]. 

PLHIV exhibited both humoral and cellular immune responses targeted specifically against SARS-CoV-2 after being administered several types of SARS-CoV-2 vaccines, such as inactivated, adenovirus vector, and mRNA vaccines [7,8,9]. The antibody responses in PLHIV increased gradually after mRNA-based vaccination [10]. However, these responses were found to be significantly lower when compared to the control group of individuals without HIV [7,11]. Low CD4^+^ T-cell numbers do not necessarily hinder cellular immune responses [7]. According to Feng et al., PLHIV who received the inactivated SARS-CoV-2 vaccination exhibited immune responses regarding binding antibodies, neutralising antibodies, and S protein-specific T-cells comparable to the HIV-negative control group [12]. Furthermore, it is worth noting that the T-cell counts of PLHIV decreased significantly following vaccination, as indicated by previous research [12]. Despite a noticeable decrease in the overall viral load following vaccination, no significant correlation was observed with CD4^+^ T-cell activation [12].

IgG specific to RBD is crucial in blocking and neutralising viral entry into target cells [2]. B cells generate neutralising antibodies in response to viral infection or vaccination [13]. They play a significant role in the humoral immune response [14]. The disease caused by SARS-CoV-2 triggers the production of NAbs, which are crucial in preventing the spike protein from binding to the ACE2 receptor in humans. This, in turn, leads to the activation of various cellular responses [7]. Furthermore, NAbs can work together with immune cells like phagocytes and natural killer (NK) cells to counteract the SARS-CoV-2 virus and prevent its escape from neutralisation [14]. NAbs provide a robust defence against symptomatic infection [2].

The progression of the SARS-CoV-2 disease is closely associated with the role of interleukins (ILs) and interferons (IFNs) [15]. IL-2 is a cytokine that plays a crucial role in regulating the function of white blood cells, particularly to activate CD4^+^ T-cells [16]. It achieves this by binding to IL-2 receptors [17]. The IL-2 level rises during infection, leading to a robust inflammatory response and cytokine storm in individuals with a weakened immune system [17]. Previous research has indicated that individuals with HIV-1 who are also infected with SARS-CoV-2 have shown the presence of serum anti-IFN-I against IFN-α subtypes, IFN-β, and IFN-ω [18]. Individuals suffering from this condition experience significant illness [18]. Furthermore, co-infected individuals exhibited heightened levels of IFNα/β and T-cell activation, indicating a significant immune dysregulation when compared to healthy individuals [19]. IFN-γ exhibited a protective role by demonstrating increased expression in convalescent patients [20]. Individuals with a compromised immune system who received IFN-γ treatment showed successful elimination of SARS-CoV-2 infection, leading to subsequent recovery from respiratory symptoms [21]. 

PLHIV are not well represented in the extensive clinical efficacy trials conducted to evaluate the effectiveness of widely used vaccinations [22]. Therefore, it is necessary to gather data on the impact of inactivated and mRNA SARS-CoV-2 immunisation. The objective of our current study is to analyse the plasma levels of anti-spike RBD SARS-CoV-2 IgG, IL-2, and IFN-γ in individuals with HIV, comparing them to healthy individuals who have received two doses of mRNA-based and inactivated SARS-CoV-2 vaccines. This study will demonstrate the beneficial effects of SARS-CoV-2 vaccination in PLHIV and offer valuable insights for developing vaccination guidelines specific to this population. 

## 2. Materials and Methods

### 2.1. Study Design and Participants 

The study site was the HIV/AIDS Seroja Clinic at Gunung Jati Regional Hospital of Cirebon, West Java, Indonesia. PLHIV were recruited as patients who have taken antiretroviral therapy (ART) regularly for at least six months and are otherwise physically healthy. The healthy control group was recruited from voluntary individuals who had attended a routine medical checkup and were HIV-negative. PLHIV cases were defined as people living with HIV who had received an mRNA-based (BNT162b2/mRNA-1273) and an inactivated SARS-CoV-2 vaccine (CoronaVac). The control group (HC) was defined as healthy individuals without HIV and any comorbidities who had received the same vaccination program. All participants were previously unvaccinated against SARS-CoV-2 and had not been infected with the virus for at least two weeks before sample collection. This study included 74 respondents, including 41 PLHIV and 33 HCs. A CD4 cell count was collected from hospital medical records before vaccination. The patients who were willing to participate voluntarily provided informed consent after being informed of the overall scope of the research. Informed consent was issued with number 047/LAIKETIK/KEPKRSGJ/XI/2021. The post-vaccination responses of each participant were tracked according to the vaccine platform they received. Blood samples were aseptically collected from each respondent on the 28th day following the first and second vaccinations [23]. Plasma was isolated for subsequent analysis after the blood was centrifuged.

### 2.2. Chemiluminescence Microparticle Immunoassay (CMIA) for RBD SARS-CoV-2 IgG

The Abbott SARS-CoV-2 IgG II Quant assay (Abbott Laboratories, Abbott Park, IL, USA) was conducted on serum samples using the Abbott Architect instrument, following the instructions provided by the manufacturer. The Architect platform used 100 μL of plasma. In the CMIA test, the SARS-CoV-2 antigen-coated paramagnetic microparticles bind to IgG antibodies attached to the virus’s spike protein in human serum and plasma samples. The chemiluminescence in relative light units (RLUs) that occurs after adding the anti-human IgG (mouse, monoclonal) acridinium-labelled conjugate can be used to measure the quantity of IgG spike protein RBD present. The strength of this response is indicated by comparing it with the IgG II calibrator/standard. Positive results are indicated by a measurement of fifty or more arbitrary units per millilitre in this test. The analytical measurement interval for the specified range is between 21 and 40,000 AU/mL, and the minimum threshold for determining positivity is set at 50 AU/mL. 

### 2.3. Analysis of IFN and IL-2 Level

The levels of IFN-γ and IL-2 were measured using a Luminex^®^ Discovery Assay kit (R&D system, Bio-Techne, Minneapolis, MN, USA) according to the manufacturer’s protocols. The measurements were performed using a Luminex Bio-Rad Analyser (Bio-Rad, Hercules, CA, USA) on plasma samples. The samples underwent centrifugation for a duration of 4 min at a speed of 1600 rpm. Then, 50 µL of IL2 standard, IFN-γ standard, and the samples were added into each well. Then, 50 µL of diluted Microparticle Cocktail was added to each well and allowed to incubate for 2 h at room temperature on a shaker set at 800 rpm and washed. The biotin antibody Cocktail was added and incubated for 1 h at room temperature on a shaker set at 800 rpm. Afterwards, the wells were washed and incubated for 2 min at room temperature on the shaker set at 800 rpm. The Luminex Bio-Rad Analyser (Bio-Rad, Hercules, CA, USA) was utilised to analyse the results.

### 2.4. Statistical Analysis

All continuous variables were checked for normality using GraphPad Prism (version 8.0, San Diego, CA, USA). The count data represented categorical variables, while medians and minimum–maximum values were used for continuous variables. The Mann–Whitney test was employed for unpaired data for two-group comparisons, while a Wilcoxon test was utilised for paired data. The correlations were assessed using non-parametric Spearman correlation tests. The figures underwent statistical analyses using R Statistical Software (v4.1.2; R Core Team 2021), utilising ggplot2, ggpubr, patchwork, and tidyverse packages. Statistical significance was determined for differences with *p*-values less than 0.05.

## 3. Results

### 3.1. PLHIV Had Lower Levels of Anti-SARS-CoV-2 RBD IgG in Response to Inactivated Vaccination but Similar Levels to HCs in Response to mRNA Vaccines

The distribution of sex and age was comparable between PLHIV and HCs. Furthermore, there was no difference in the treatment time of antiretroviral therapy (ART) and CD4 cell counts among PLHIV who were administered mRNA and inactivated vaccines (Table 1). There was no difference in the sex proportion, age, time of ART, and CD4 cell counts between PLHIV in mRNA and inactivated vaccine groups. 

### 3.2. PLHIV Who Received the SARS-CoV-2 mRNA Vaccine Had High Neutralising Antibody Levels after the First Vaccination

The SARS-CoV-2 reactive IgG was higher in HIV patients who received the mRNA vaccine than those who received the inactivated vaccine. The SARS-CoV-2 reactive antibody to RBD was higher in HCs than in PLHIV receiving the inactivated vaccine after the first dose. The mRNA vaccine group had higher anti-RBD IgG than the inactivated vaccine group in PLHIV and HCs (Figure 1A,B). However, it was observed that the levels of IgG anti-RBD in HIV patients who received the mRNA vaccine decreased after the second dose, with no significant difference between the two time points (Figure 1C). By contrast, PLHIV who received the inactivated vaccine had an increased IgG-specific RBD after the second dose (Figure 1C). There was no difference in HCs IgG-specific RBD after the first and second dose of inactivated and mRNA vaccines (Figure 1D). 

### 3.3. High Levels of IL-2 and IFN-γ Were Present in PLHIV with the First Dose of the Inactivated SARS-CoV-2 Vaccine and Then Decreased on the Second Dose of Vaccination

The levels of IL-2 and IFN-γ in circulation were assessed during the two vaccination periods. The levels of IL-2 showed an increase after the initial dose of the inactivated vaccine, with no discernible difference observed between the groups of healthy individuals who received either the inactivated or mRNA vaccines. For PLHIV who received the inactivated vaccine, the level of IL-2 was found to be higher compared to those who received the mRNA vaccine (Figure 2A). There were no significant differences in IFN-γ levels among the groups following the initial vaccination with inactivated and mRNA vaccines (Figure 2B). The levels of IL-2 and IFN-γ in the second dose of both vaccination platforms were similar across all groups (see Figure 2B,D). For PLHIV who received the inactivated vaccine, the levels of IFN-γ were slightly higher in the second dose compared to PLHIV who received the mRNA vaccine (*p* = 0.08). The levels of IFN-γ in people living with HIV who received the second dose of the inactivated vaccine were found to be higher compared to healthy individuals who also received the inactivated vaccine (Figure 2D). 

### 3.4. IL-2 and IFN-γ Levels Were Strongly Linked to Anti-RBD SARS-CoV-2 IgG in the Second Dose of Vaccination, Regardless of CD4 Cell Counts in PLHIV Receiving mRNA Vaccination

PLHIV demonstrated increased levels of IgG following the administration of a second dose of the inactivated vaccine and a first dose of the mRNA vaccine. The study examined the correlation between CD4 cell counts and IgG levels before and after vaccination. No notable correlation was observed between the inactivated and mRNA vaccination groups (Figure 3A,B). The levels of IgG showed a clear, direct correlation with the cytokines IL-2 (Figure 3C) and IFN-γ (Figure 3D), specifically in individuals with HIV who received the mRNA vaccine. On the other hand, there was a negative correlation observed between IgG-specific RBD and the IFN-γ found in HCs who received the mRNA vaccine (Figure 3C). 

## 4. Discussion

The study revealed a notable disparity in the levels of reactive IgG-specific RBD SARS-CoV-2 between PLHIV who received mRNA-based vaccines and those who received inactivated vaccines. The levels of SARS-CoV-2 RBD IgG after the second dose showed a direct correlation between IL-2 and IFN-γ levels in mRNA-vaccinated PLHIV. On the other hand, the inactivated vaccine platform showed a higher proportion of IL-2 and IFN-γ in PLHIV after the first dose. 

High SARS-CoV-2 RBD IgG levels of PLHIV may be linked to the mRNA-based vaccine’s ability to effectively utilise the host cells’ protein machinery to convert mRNA into a targeted antigen to elicit a robust immune response [13,24]. This was consistent with previous reports that SARS-CoV-2 mRNA-based vaccines elicit more robust cell-mediated immunity (CMI) than SARS-CoV-2 inactivated vaccines [25]. In addition, our study found that inactivated SARS-CoV-2 vaccination may need a second to a third dose to induce two times the NAb levels measured through surrogate viral neutralisation assay (sVNT) [26]. Typically, NAbs are used to measure antibodies against the spike protein. One advantage of the inactivated vaccine is that it measures antibodies against other proteins, such as the robust nucleocapsid-specific IgG [27]. 

Stimulated peripheral mononuclear cells (PBMCs) from PLHIV exhibited comparable levels of IFN-γ to several immunocompromised groups and healthy donors, despite lower levels of SARS-CoV-2 RBD IgG compared to healthy donors who received inactivated vaccines [28]. The levels of IL-2 showed a direct association with IFN-γ, indicating their potential to predict a protective response against viral infection [29,30]. Inactivated vaccines have been found to elicit heterologous T-cell responses against various components of SARS-CoV-2, including spike, nucleocapsid, and membrane proteins [31]. PLHIV who were administered a booster dose of the SARS-CoV-2 Monovalent Omicron XBB mRNA vaccine showed a notable rise in NAbs three months later [32]. 

The findings of our study revealed an increase in the levels of IFN-γ and IL-2 in PLHIV who received inactivated vaccines, as opposed to those who received mRNA vaccines. Nevertheless, the strong relationship between IFN-γ/IL-2 and SARS-CoV-2 RBD IgG in the mRNA group suggests a significant role of spike-specific antibodies in viral immunity. On the other hand, inactivated vaccines may elicit a more comprehensive cellular response to various SARS-CoV-2 strains. 

PLHIV are more likely to contract other infections, which cause lymphocyte stimulation and the generation of pro-inflammatory cytokines [33]. As a result, more HIV is replicated at an increased pace, and most CD4^+^ T-cells are lost; this may result in severe infection [33]. Regarding SARS-CoV-2 infection in PLHIV, CD4 T-cell counts above 500 cells/µL are associated with responsive humoral and cellular immune response to vaccination [22]. The levels of immunological response decreased when stratified through CD4 levels ranging from 200 to 500 cells/µL or less than 200 cells/µL [34]. However, it was observed that the CD4 cell counts were above 200 cells/µL in this study. PLHIV exhibited elevated levels of neutralising antibody responses following vaccination despite the CD4 counts. It is worth noting that all our patients living with HIV have been receiving antiretroviral therapy (ART) for a minimum of six months. 

Our study is limited by the small sample size. Therefore, it may not accurately reflect the population of PLHIV who have different antiretroviral treatment statuses and CD4 cell counts. The cytokines analysed in our study were obtained from plasma rather than stimulated peripheral blood mononuclear cells (PBMCs). Additional research should explore the plasma and cellular cytokine levels in PLHIV who have been exposed to various SARS-CoV-2 proteins. In this study, we did not analyse the ART regimen. Types of ART could inhibit SARS-CoV-2 infection, and protease inhibitors could also have an inhibitory effect on SARS-CoV-2 infections in PLHIV [35]. 

## 5. Conclusions

Our study concludes that mRNA-based immunisation effectively generated strong antibodies reactive to SARS-CoV-2 in PLHIV who are undergoing ART, regardless of their CD4 cell counts. The inactivated vaccine may provide cellular immunity by increasing the levels of IL-2 and IFN-γ in PLHIV. The findings demonstrated a positive immune response to vaccination in PLHIV, supporting the importance of prioritising SARS-CoV-2 vaccination in this population.

## Figures and Tables

**Figure 1 biomedicines-12-02115-f001:**
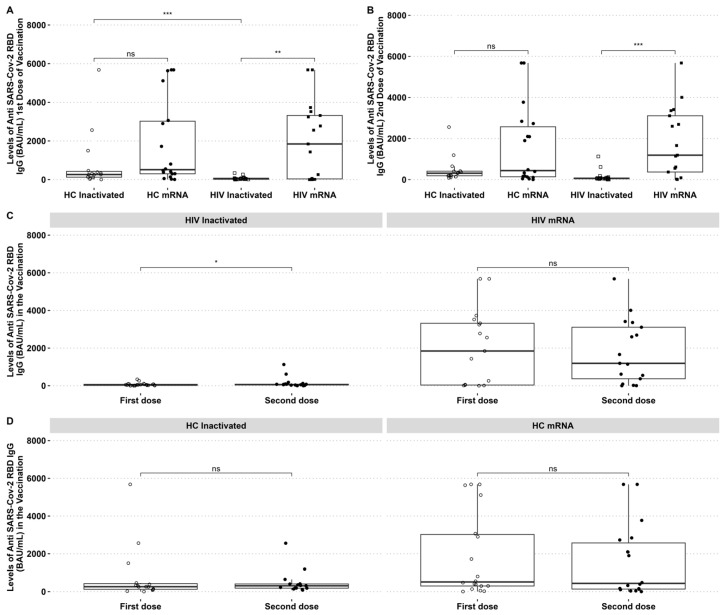
PLHIV and HCs who received mRNA vaccines had considerably greater IgG anti-RBD SARS-CoV-2 antibody levels than those who received inactivated vaccines. (**A**) Comparative analysis of IgG levels between healthy individuals and PLHIV for the first and (**B**) second doses. (**C**) Changes in IgG levels in inactivated and mRNA PLHIV before and after vaccination. (**D**) IgG levels of HCs before and after vaccination. * *p*-value ≤ 0.05, ** *p*-value ≤ 0.01, *** *p*-value ≤ 0.001, and ns = not significantly different.

**Figure 2 biomedicines-12-02115-f002:**
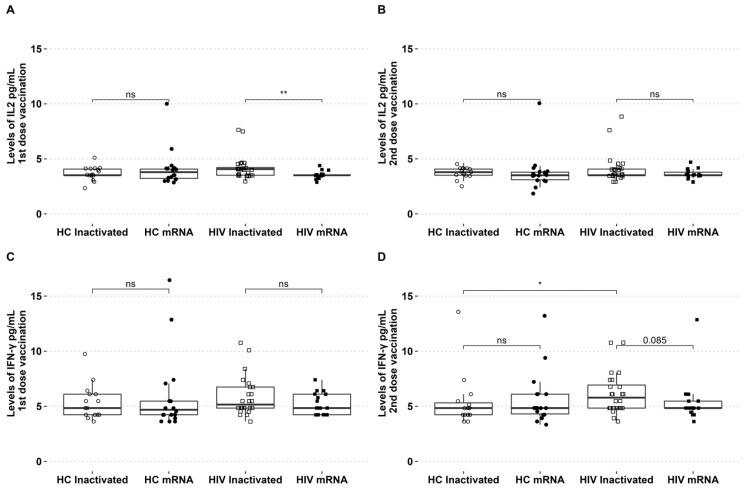
PLHIV receiving inactivated vaccinations exhibited high IL-2 after the first dose and high IFN-γ after the second dose. (**A**) Levels of IL-2 after the first and (**B**) second vaccination. (**C**) IFN-γ levels of first and (**D**) second vaccination in HCs and PLHIV receiving inactivated or mRNA SARS-CoV-2 vaccination. * *p*-value ≤ 0.05, ** *p*-value ≤ 0.01, and ns = not significantly different.

**Figure 3 biomedicines-12-02115-f003:**
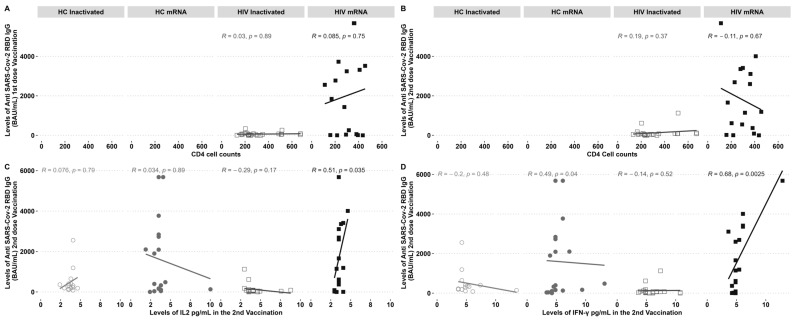
Anti-RBD IgG SARS-CoV-2 antibodies and elevated IL-2 and IFN-γ levels were found in PLHIV who underwent mRNA immunisation after the second dosage. Anti-RBD IgG SARS-CoV-2 antibodies were not affected by CD4 cell counts. (**A**) Correlation of CD4 cell counts pre-vaccination with an IgG after the first and (**B**) second vaccination; (**C**) IL2 and (**D**) IFN-γ correlation with an IgG post-second dose vaccination. *R* = correlation coefficient and *p *= the *p*-value.

**Table 1 biomedicines-12-02115-t001:** Demographic data of healthy controls and HIV patients receiving two doses of SARS-CoV-2 vaccination.

	HC		PLHIV						
	mRNA Vaccine A	Inactivated Vaccine B	mRNA Vaccine C	Inactivated Vaccine D	A–D	A vs. B	C vs. D	A vs. C	B vs. D
n	18	15	17	24	-	-	-	-	-
Sex									
Male	11	8	11	10	0.45 ^a^	-	-	-	-
Female	7	7	6	14		-	-	-	-
Age	31 (24–50)	41 (19–72)	35 (25–44)	38 (18–50)	-	0.1 ^b^	0.21 ^b^	0.38 ^b^	0.32 ^b^
Time on ART (years)	**-**	**-**	2 (1–4)	3 (1–4)		-	0.52 ^b^	-	-
CD4 cell counts	-	-	299 (112–456)	251 (131–673)	-	-	0.89 ^b^	-	-

Data are presented as median (range). ^a^ *p*-value: Chi-square test of male and female proportions among groups; ^b^
*p*-value: Mann–Whitney test.

## Data Availability

The raw data supporting the conclusions of this article will be made available by the authors on request.

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
