# Peer review of "Evaluation of the Effect of mRNA and Inactivated SARS-CoV-2 Vaccines on the Levels of Cytokines IL-2, IFN-γ, and Anti-RBD Spike SARS-CoV-2 Antibodies in People Living with HIV (PLHIV)"

_biomedicines, 2024, doi:10.3390/biomedicines12092115_

Round 1
Reviewer 1 Report
Comments and Suggestions for Authors
The authors try to study the effect of mRNA and inactivated SARS-CoV-2 vaccines on the levels of cytokines IL-2, IFN-γ, and Anti-RBD Spike SARS-CoV-2 antibody in people living with HIV.
The subject seems to e very interesting and could provide interesting information that contribute in the understanding of the SARS-CoV-2 vaccines. However, the manuscripts in its methodology and structure has shown multiple gaps that have affected its quality.
1. The main concern is related to the methodology. In fact, the authors did not provide how were the participants selected (for cases and control). They did not provide practically any information regarding these participants and the relation between the cases and the controls. In addition, the authors did not provide a full description for the laboratory methods used and they just provide that the they used the manufacturer's instructions (the same remark for the statistical analysis).
2. The second concern is related to the way of writing in all parts of the manuscript which lacks a certain rigor in all its parts (from the introduction to the conclusion) where the hypothesis was not defined and the conclusion did not support and is not in accordance with what was done (see comment below).
3. The (other) detailed comments for each part are :
-Title: reformulate : it may be "evaluation of the effect… "
- Abstract:
The introduction is composed of 3 sentences that has not been correlated. Reformulate
Line 34: "the level of…in…. " of what?
Conclusion: It has no relation with the results. You should provide the most important results.
- Introduction:
The introduction is composed of multiple no correlated paragraphs and the effect of summarizing in not seen. It has not allowed to define the hypothesis of the work. You should reformulate, summarize it and correlate between the different ideas.
Line 55: It is currently recommended? Or just in the past? (were)
Line 59: "The condition…duration". Are you talking about the virus or about the disease?
Line 83: Is this reference 12 or 21? Correct.
Line 90: "The study…COVID-19". It may be " A study…". Provide the complete definition of COVID-19.
- Methods:
Line 118: Provide the description of the hospital
Line 120: delete 19 after " SARS-CoV-2"
Line 120-122: PLHIV …adherence". Are you talking about your subjects or about all PLHIV (who were probably excluded)? Why did you provide this information (excellent adherence) ? we are in the methods not in the discussion.
How did you select these individuals ? why this number for cases and controls? The relation between them? What is the duration between the first and the second dose? Why 28 days? .....
Line 134-142 and 144-146: provide more details regarding the used techniques
Line 142: The correlation of what? How did you calculate the correlation? Are you sure the it can be tested here?
- Results :
The content of the table is repeated in the figures. You should provide in the table just what cannot be shown in the figures. You should indicate the table multiple times.
Lines 156-159, 166-168, 178-179, 194-195, 203-205, 220-222: you should avoid to use the introductory sentences (of the methods) and discussing your results in this part. You should also avoid to indicate the table in each sentence (as illustrated, as shown…..).
Improve the quality of the table and separate between the variables
- Discussion:
Try to reformulate the discussion by deleing the unnecessary details (i.e: the future direction of …SARS-CoV-2 variants) and add more related references. You should also correlate between your ideas.
Line 243:" …need a second or a third dose.." how did you find this ? have you included the third dose in your work?
Line 245: add a reference "Typically….protein".
Line 290: what do you mean b "optimal"? what was your scale?
Line 193: "evaluating" may be "elevating"?
Line 293-294: it is important..counts" what has this sentence in relation with your results?
You should reformulate the conclusion by providing the most important results that could answer to the hypothesis of the work.
Comments on the Quality of English Language
Minor editing rquired
Author Response
we have road answer on the attached document. Thank you

Reviewer 2 Report
Comments and Suggestions for Authors
The authors determined anti-RBD IgG, IL-2 and IFN- γ titers in 74 participants received SARS-CoV-2 vaccines to compare immune responses individuals living with or without HIV. Although the sample numbers are small, this is an important area to analyze. Specific comments follow.
Major points:
1. Line 167: I’m not sure why the authors say “it is advised to administer a minimum of two doses”. Is there any significant difference in terms of protection in 45.5 vs 60 BAU/mL?
2. Figure 1: Please add similar graph to 1C for NC group for comparison as 1D. 1C title should be PLHIV but not HIV for consistency.
3. Line 291: Do the authors have data supporting sustainability of the neutralizing antibody?
4. Lines 291 and 294: These are contradicting description. Does CD4 cell count important or not?
Minor points:
1. Table 1: Please explain “HNC”.
2. Line 212: Abbreviation should be used. Please delete “Individuals living with HIV” and carefully check all other abbreviation used such as “ART”, and “PBMC”.
3. Line 267: “cells” is missing after “200”.
Please include more recent publicaions such as;
Immunogenicity of the Monovalent Omicron XBB.1.5-Adapted BNT162b2 COVID-19 Vaccine in People Living with HIV (PLWH).
Cherneha M, Zydek I, Braß P, Korth J, Jansen S, Esser S, Karsten CB, Meyer F, Kraiselburd I, Dittmer U, Lindemann M, Horn PA, Witzke O, Thümmler L, Krawczyk A.
Vaccines (Basel). 2024 Jul 17;12(7):785. doi: 10.3390/vaccines12070785.
Interleukin-2-mediated CD4 T-cell activation correlates highly with effective serological and T-cell responses to SARS-CoV-2 vaccination in people living with HIV.
Gupta A, Righi E, Konnova A, Sciammarella C, Spiteri G, Van Averbeke V, Berkell M, Hotterbeekx A, Sartor A, Mirandola M, Malhotra-Kumar S, Azzini AM, Pezzani D, Monaco MGL, Vanham G, Porru S, Tacconelli E, Kumar-Singh S.
J Med Virol. 2024 Aug;96(8):e29820. doi: 10.1002/jmv.29820.
SARS-CoV-2-specific T-cell responses are induced in people living with human immunodeficiency virus after booster vaccination.
Wang X, Li Y, Jin J, Chai X, Ma Z, Duan J, Zhang G, Huang T, Zhang X, Zhang T, Wu H, Cao Y, Su B.
Chin Med J (Engl). 2024 Jul 18. doi: 10.1097/CM9.0000000000003176.
COVID-19 Vaccines and COVID-19 in People Living with HIV.
Karaşın MF, Bayraktar Z, Toygar-Deniz M, Akhan S, Özdemir MK.
Infect Dis Clin Microbiol. 2024 Jun 28;6(2):78-82. doi: 10.36519/idcm.2024.271.
Neutralizing antibody responses assessment after vaccination in people living with HIV using a surrogate neutralization assay.
Batchi-Bouyou AL, Djontu JC, Ingoba LL, Mougany JS, Mouzinga FH, Dollon Mbama Ntabi J, Kouikani FY, Christ Massamba Ndala A, Diafouka-Kietela S, Ampa R, Ntoumi F.
BMC Immunol. 2024 Jul 10;25(1):43. doi: 10.1186/s12865-024-00625-z.
Author Response
we have wrote the answers on the attached document. thank you

Round 2
Reviewer 1 Report
Comments and Suggestions for Authors
I would like to thank the authors for their efforts to improve the quality of the manuscript. However, it still still lacks a certain rigor and mastery. I have some comments that interplead me:
The introduction of the abstract is very simplistic and did not define he hypothesis of the work (why has the study been conducted?).
Line 20: delete the term "platform".
Line 35: administering a two dose is essential ….". Reformulate: it is just your observation and cannot be a recommendation.
Add your recommendations after the conclusion.
Introduction:
Line 43: delete the expression "HIV infection" in the second sentence to avoid repetition.
Line 53 "People living….mRNA vaccine". What did this sentence mean?
Line 63: delete "as indicated by previous research" since the reference was cited before
Lines 89-91: reformulate this sentence.
Materials and Methods
Lines 101 and 105: "Gunung Jati Regional Hospital HIV/AIDS Seroja Clinic", " Gunung Jati Hospital". Are they the same hospital? Why did you repeat it? Delete the repetition.
Lines: "PLHIV are defined…". PLHIV are defined as people living with HIV and cannot be defined otherwise. It may be : "cases are as PLHIV….". The same for the controls.
Line 118: you should provide the reference for 28 days.
Results:
You deleted the content of the table but you did not provide the results of the statistical analysis for the figure, you should complete the comparison of line 169-177 with the results of the statistical analysis provided in the first table (first version). You should provide also these results in the figures.
In addition in figure 1A-B: you compared just between HIV patients. You should show the results for the controls.
Line 63: received may be receiving
Table 1 : you can also compare the % ofindividuals (for sex for example) using Chi squared or Fisher tests to see if is there a statistical difference.
Line 165: Delete "Bold: p-value ≤ 0.05. italic: p-value <0.1"
Lines 169-170: delete the first sentence
For all the figures: reformulate the titles to be clear and descriptive of the content.
Line 192: delete "Figure 2A"
Line 204-205: delete the sentence: "The study…vaccination".
The conclusion is very simplistic. It should be reformulated. You should also add recommendations
Comments on the Quality of English Language-
Author Response
Dear Reviewers,
We thank the reviewers for the valuable suggestions to our manuscript. In this document, we have included a confirmation, explanation, and revisions pertaining to the comments for 2nd round of the review process. Revised sections of the manuscripts were marked with blue colour.
We believe that this input dramatically enhances the quality of our manuscript.
Kind Regards,
Amanah

Reviewer 2 Report
Comments and Suggestions for Authors
Once you define an abbreviation, please use only the abbreviation for later mentions. Do not alternate between spelling out the term and abbreviating it. So, please write “people living with HIV (PLHIV)” in Line 21 then you should use the shortened form rather than the full term for later mentions such as lines 23, 36, 38 etc.
Author Response

(The authors gave the same response as above.)

Round 3
Reviewer 1 Report
Comments and Suggestions for Authors
I congratulate the authors for their efforts. I have juste some"minor" comments:
In your figures, I suggest to reformulate the titles that should be informative and concise (you can add after this the despcription that you provided as a title).
- In table 1 add a title as "p value" in the first line
also when compaing with Ch-squared test, you should the same thing as for the other test (A-B, AC.....), you should provide the value for each parametter. In addition, the provided description of "a" and "b" is not correct. You shold assocate them with the p value not with the variable (AB, AC, BC, ...).
Example: 0.1b, 0.21b .....should e associated with b (Mann-Whitney test) while the results of Chi-squared for AB, AC...should be associated with a.
Author Response
We thank the reviewers for the valuable suggestions on every detail of our manuscript. In this document, we have included the revisions pertaining to the comments for 3rd round of the review process.
Revised sections of the manuscripts were marked with blue colour.
We believe that this input dramatically enhances the quality of our manuscript to be considered for publication.
Kind Regards,
Amanah
